# Novel Hemocompatible Imine Compounds as Alternatives for Antimicrobial Therapy in Pharmaceutical Application

**Mohammad A. Altamimi** [1,*]**, Afzal Hussain** [1] **, Sultan Alshehri** [1,2]**, Syed Sarim Imam** [1]**, Abdulmalik Alnami** [1] **and Ahmed Bari** [3,4]

[1] Department of Pharmaceutics, College of Pharmacy, King Saud University, Riyadh 11451, Saudi Arabia; amohammed2@ksu.edu.sa (A.H.); salsheri1@ksu.edu.sa (S.A.); simam@ksu.edu.sa (S.S.I.); 436100057@student.ksu.edu.sa (A.A.)

[2] College of Pharmacy, Almaarefa University, Riyadh 11451, Saudi Arabia

[3] Central Laboratory, College of Pharmacy, King Saud University, Riyadh 11451, Saudi Arabia; abari@ksu.edu.sa

[4] Department of Pharmaceutical Chemistry, College of Pharmacy, King Saud University, Riyadh 11451, Saudi Arabia

\* Correspondence: maltamimi@ksu.edu.sa; Tel.: +96-61-1467-3572

**Abstract:** The aim of this study was to synthesize, characterize, and evaluate neoteric imine compounds for antimicrobial activity and hemocompatibility. Four compounds were synthesized using 3-thiophene carboxaldehyde, ethanol, amine, and acetic acid. The compounds were characterized using nuclear magnetic resonance (NMR) spectroscopy, Fourier transform infrared (FTIR) spectroscopy, differential scanning calorimetry (DSC), and powder X-ray diffraction (PXRD). A solubility study was conducted with various solvents and surfactants at 40 °C. An in vitro antimicrobial assay was performed against bacterial and fungal strains to determine the zone of inhibition and minimum inhibitory concentrations. Finally, an in vitro hemolysis study was conducted using rat erythrocytes. The structure of the compounds was confirmed by NMR, FTIR corroborated their functional group attributes, DSC determined their enthalpies of fusion and fusion temperatures, and PXRD confirmed their crystalline nature. These compounds were water-insoluble but soluble in chloroform, with a maximum solubility of ~80 mg/mL. The antimicrobial assay suggested that two of the products exerted potent activities against *C. albicans* and several bacterial strains. Finally, hemolysis analysis excluded the possibility of hemolysis at the assessed concentrations. In conclusion, two of the novel imine compounds showed promise as antimicrobial agents to control local and systemic microbial infections in a suitable dosage form.

**Keywords:** novel imine compounds; characterization; in vitro antimicrobial assay; in vitro hemolysis; local and systemic infection control

## 1. Introduction

Bacteria and fungi are the most prevalent microbes on earth, a great variety of which coexists as versatile commensals with mammals. These interactions can be both beneficial and detrimental to human health, depending on numerous microbial- and host-related factors. Since the discovery of penicillin, numerous additional antibiotics have been identified in bacteria and fungi, while others have been synthesized with improved effectiveness and safety. However, the exhaustive use of antibiotics and antifungal drugs has led to the emergence of drug resistance over time, thereby negating their efficacy. The emergence of resistant strains has greatly challenged the clinical application of several drugs to

control bacterial and fungal infections. Moreover, other factors (climate, ethnicity, genetics, habitat, diet, and diversity) can also alter host susceptibility to opportunistic pathogens. Notably, concurrent with the increased incidence of resistance to antimicrobial agents, there has been a corresponding decrease in the discovery and development of new drug candidates [1]. These observations highlight the urgent need to develop new drugs with improved therapeutic efficacy and acceptable safety.

Chemically, an imine or azomethine (also known as a Schiff base) is a "nitrogen" analog of an aldehyde or ketone in which a carbonyl functional group (C=O) is replaced with an imine or azomethine group [2]. da Silva et al. reviewed and highlighted various examples of imine-containing compounds possessing antibacterial, antifungal, antiviral, and antimalarial potential [2]. Imines are the best known and most frequently used intermediates in the synthesis of nitrogen-containing organic analogs, particularly those with bioactivity potential. Imines are versatile precursors in organic synthesis, and numerous publications have covered their use in therapeutic or biological applications, such as in the development of potential drug candidates, diagnostic probes, and analytical tools [2–5]. They are present in various natural, semi-synthetic, and synthetic compounds and have been shown to be essential for their biological activities. Some promising imine-based natural and synthetic molecules are shown in Figure 1 [6,7].

**A** Ancistrocladidine
*Antimalarial activity*

**B** N-salicylidene-2-hydroxyaniline
*Antibacterial activity*

**Figure 1.** Structures of commercially available imine-based bioactive molecules. (**A**) Ancistrocladidine Antimalarial activity, (**B**) N-salicylidene-2-hydroxyaniline.

Imines and the reactions involving the imine functional group are the backbone of synthetic chemistry and the synthesis of potentially biologically active molecules. The hydrolysis of Schiff bases is involved in several enzyme-mediated processes, especially in "Sommerlet" and "Gattermann aldehyde synthesis" [8,9]. They are macrocyclic ligands that contain nitrogen as the donor and are often found in polydentate in coordination compounds [10,11]. Metal complexes of Schiff bases are also an important tool in the fight against bacteria and viruses and have proven more active than their parent molecules. In an aqueous solution of imines, water is added across the imine bond and exists in equilibrium with the carbinolamine form. Imines derived from acyclic amines are usually unstable and undergo nonenzymatic hydrolysis to the corresponding amine and aldehyde metabolites [12,13]. Additional examples of drugs that form stable iminium ions as metabolites include 1-methyl-4-phenyl-1,2,3,6-tetrahydropyridine, a neurotoxin derivative of haloperidol. Both of these drugs form positively charged iminium metabolites that are responsible for their neurotoxicity [13,14].

Thiophene-3-carboxaldehyde was selected in the present study based on its established biological activity. It has been used as an intermediate in the synthesis of various biologically distinct molecules. Most recently, this carboxaldehyde was used to synthesize pyrazoline as an inhibitor of catechol-O-methyltransferase (COMT) and monoamine oxidase (MAO) [15]. Additionally, thiophene-3-carboxaldehyde has also been used to synthesize piperidine analogs for the treatment of Parkinson's disease, thereby confirming its versatility as a pharmacophore [16].

Considering the diverse biological activities of imines and chemistry, we describe the synthesis of new analogs derived from the reaction of primary imines with thiophene-3-carboxaldehyde.

Four imines (**AM-3**, **AM-5**, **AM-7**, and **AM-8**) were synthesized aiming to produce compounds with significant antibacterial and antifungal potential as alternatives for use in chemotherapy. The structures of the compounds were confirmed using nuclear magnetic resonance (NMR) spectroscopy, while the attributes of the functional groups were verified by Fourier transform infrared (FTIR) spectroscopy. Moreover, differential scanning calorimetry (DSC) was used to determine the melting point and fusion enthalpy of the synthesized compounds. Powder X-ray diffraction (PXRD) and scanning electron microscopy (SEM) corroborated the solid-state characteristics of the compounds. It was essential that the compounds were solubilized in organic solvents and surfactants for further analysis. The compounds were assayed for their antibacterial and antifungal properties in vitro. Finally, these compounds were assessed for their preliminary safety (hemocompatibility) using rat erythrocytes (hemolysis study). These compounds may be suitable for topical, transdermal, and oral delivery in an appropriate formulation after the completion of preclinical studies.

## 2. Materials and Methods

### 2.1. Materials

All the reagents were purchased from Sigma–Aldrich Chemical Co. The solvents used in this study (ethanol, dimethyl sulfoxide, chloroform, and methanol), were of high-performance liquid chromatography (HPLC) grade and were procured from Sigma–Aldrich Chemical Co. DMSO-$d_6$ with 1% tetramethylsilane (TMS) was purchased from Cambridge Isotopes, Ltd. and used for NMR spectroscopy. Labrasol (LAB) and Cremophor-EL were kind gifts from Gattefosse (St-Priest, Cedex, France) and BASF (Ludwigshafen, Germany), respectively. Miglyol 829 and Tween 80 were obtained from Sasol, Ltd., South Africa, and Sigma–Aldrich, respectively. *Staphylococcus aureus* (ATCC-29213), *Bacillus subtilis* (ATCC-10400), *Escherichia coli* (ATCC-25922), *Acinetobacter baumannii* (clinical 20 KSA), *Enterococcus faecalis* (ATCC-29212), nonvirulent *Mycobacterium smegmatis* (ATCC-35797), *Candida albicans* (ATCC-10231), and *Aspergillus Niger* (ATCC-16404), were obtained from the Department of Microbiology, College of Pharmacy, King Saud University, Riyadh, Saudi Arabia. Positive controls such as ketoconazole, isoniazid, and rifampicin were obtained from Unicure India Pvt Ltd. (New Delhi, India).

### 2.2. Methods

#### 2.2.1. Chemical Synthesis

An alcoholic solution of 10 mmol 3-thiophene carboxaldehyde (20 mL in absolute ethanol) was mixed with the selected amine (11 mmol) followed by the addition of acetic acid (5 drops). The reaction mixture was stirred at room temperature for 30 min followed by reflux for 3 h. After completion of the reaction, the solvent was evaporated in a vacuum, and ice was added. The obtained compounds were purified as per previously described methods, with slight modifications [17–20]. The obtained precipitates were filtered through a polycarbonate membrane filter (compatible and water-insoluble) to remove the solvent. Then, the precipitate was repeatedly washed with distilled water for the complete removal of the solvents, followed by evaporation, under vacuum, using a rotary. The dried forms of the compounds were of high yield value (Figures 2 and 3).

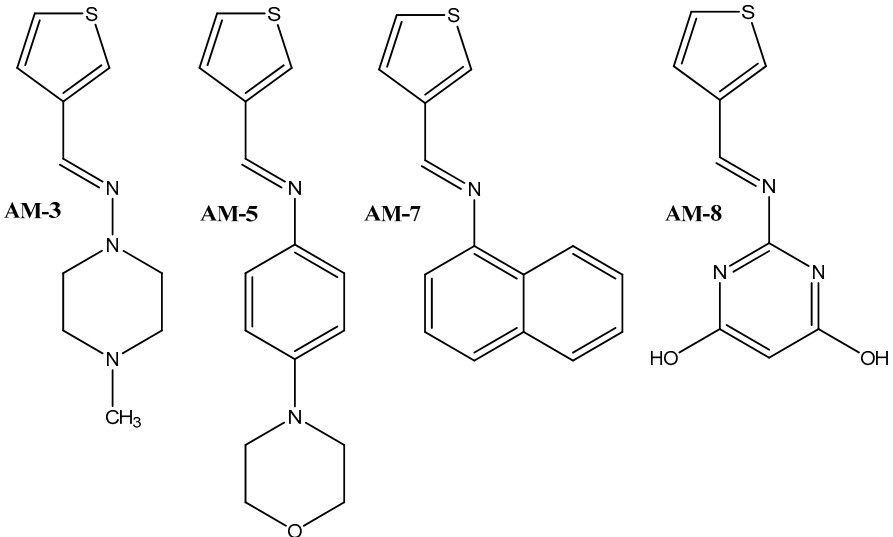

**AM-3**: $R^1$ = 1-amino-4-methylpiperazine
**AM-5**: $R^1$ = 4-morpholinoaniline
**AM-7**: $R^1$ = 1-naphthylamine
**AM-8**: $R^1$ = 2-amino-4,5-dihydroxypyrimidine

**Figure 2.** Schematic of the one-step synthesis of the imine compounds (**AM-3**, **AM-5**, **AM-7**, and **AM-8**) in the presence of ethanol and acetic acid (AcOH).

**Figure 3.** The chemical structure of 4-methyl-*N*-(thiophen-2-ylmethylene)piperazin-1-amine (AM-3 [$C_{10}H_{15}N_3S$]; yield: 72%; mass: 209.10 g/Mol); 4-morpholino-*N*-(thiophen-2-ylmethylene)aniline (AM-5 [$C_{15}H_{16}N_2OS$]); yield: 79%; mass: 272.10 g/Mol). *N*-(thiophen-2-ylmethylene)naphthalen-1-amine (AM-7 [$C_{15}H_{11}NS$]; yield: 73%; mass: 237.06 g/Mol); 2-([thiophen-2-ylmethylene]amino) pyrimidine-4,6-diol (AM-8 [$C_9H_7N_3O_2S$]; yield: 70%; mass: 221.03 g/Mol).

2.2.2. Nuclear Magnetic Resonance Spectroscopy

Nuclear magnetic resonance spectroscopy ($H^1$ NMR [700 MHz]; Bruker NMR; Topspin software v.3.2) was used to elucidate the structure of the synthesized compounds. For the synthesis of compounds **AM-3, AM-5, AM-7,** and **AM-8,** thiophene-3-carboxaldehyde was treated with selected primary amines in refluxing ethanol with a few drops of acetic acid. The disappearance of the formyl proton and the presence of a singlet for CH at around 8 ppm confirmed the postulated structures. In the **AM-3** and **AM-5** compounds, the appearance of morpholino and piperazine protons between 2 and 3 ppm and a broad peak in the downfield region for OH in the **AM-8** compound confirmed the postulated structures. The synthesized compounds were washed several times with water and analytically pure compounds were obtained for further analysis [21,22].

### 2.2.3. Fourier Transform Infrared Spectroscopy

This technique is used to identify characteristic functional groups as "fingerprints" of the synthesized compounds. The principle is based on bond vibration and bond stretching under the influence of infrared energy. The characteristic peaks and spectral patterns ensure compound purity by identifying specific functional groups. Any significant deviation of the peaks and spectral patterns from standard references when using Alpha Bruker Platinum-ATR is indicative of possible compound impurities, chemical interactions, or degradation. The lens was cleaned using distilled water and wiped before testing, and internal calibration was ensured before spectral measurements were made. The spectra were recorded at a resolution of 4 cm$^{-1}$ over the wavenumber range of 4000 to 400 cm$^{-1}$, with 24 scans per sample [23,24].

### 2.2.4. Differential Scanning Calorimetry

The melting points of the four novel compounds—**AM-3**, **AM-5**, **AM-7,** and **AM-8**—were determined using DSC (DSC-8000, PerkinElmer, Waltham, MA, USA) equipped with Pyris Manager software. A precisely weighed sample (3 mg) was placed in a new pan followed by crimping to achieve a hermetically sealed state. Later, the sample was placed in a furnace chamber along with a blank crimped pan that was used as a reference. All the samples were processed by heating from 30 to 350 °C at a rate of 10 °C/min under constant nitrogen flow (40 mL/min) [25,26].

### 2.2.5. Powder X-ray Diffractometry

All the synthesized solid powders were subjected to PXRD (Ultima IV diffractometer, Rigaku Inc. Tokyo, Japan) to determine their diffraction patterns (solid-state behavior) using a previously described method [27,28]. The samples were thinly smeared on a glass coverslip and scanned from (2 theta) 3° to 60° with an angular scanning rate of 0.5° per min. To analyze the crystalline characteristics, data were collected using primary monochromatic radiation (Cu K$\alpha$1, $\lambda$ = 1.54 Å). Scanning was carried out at an operating voltage of 40 kV and a working current of 40 mA (step size 0.02°, 1 s per step).

### 2.2.6. Morphological Analysis Using Scanning Electron Microscopy

SEM (Model FEI, Inspect-S50, Moravia, Czech Republic) was used to assess the morphology of the synthesized compounds [29]. The samples were fixed to the brass stub with double adhesive tape and then coated with gold using an ion sputter to render them conductive for scanning at ambient temperature. Scanning was conducted at a working voltage of 10 kV.

The chemical composition (elemental assessment) and percent composition of the compounds was evaluated using energy-dispersive X-ray spectroscopy (EDX) coupled with SEM (EDX SEM) [30].

### 2.2.7. Solubility Analysis

The solubility of the compounds was quantified in various solvents (ethanol, chloroform, water, and DMSO) and surfactants (LAB, Tween 80, miglyol 829, and Cremophor-EL) solutions [31,32]. When required, compounds were serially diluted with 2 mL of each solvent using a stopper glass vial (10 mL). The mixture was vortexed for 10 min and then placed on a shaking water bath set at 40 °C for 48 h. Each compound was continuously added into the explored solvent until saturation, which was confirmed by the formation of visible precipitation. The vial was then centrifuged to remove undissolved materials. The amount of dissolved compound was estimated using a UV-Vis spectrophotometer (Shimadzu, U-1800 Spectrophotometer, Tokyo, Japan). These compounds were diluted using 5% LAB. The $\lambda_{max}$ values of **AM-3, AM-5, AM-7,** and **AM-8** at 302, 253, 289, and 327 nm, respectively, were determined using a spectrophotometer.

### 2.2.8. In Vitro Antimicrobial Activities

Zone of Inhibition

As all the compounds were water-insoluble, they were instead solubilized in LAB aqueous solution (5% *v/v*). Stock solutions (1 mg/mL) of **AM-3, AM-5, AM-7,** and **AM-8** were prepared in the same solution. Nutrient broth medium was used to revive the bacterial cultures, and each strain (*S. aureus*, *B. subtilis*, *E. coli*, *A. baumannii*, *E. faecalis*, and nonvirulent *M. smegmatis*) was grown to an optical density ($OD_{600}$) of 0.6 to obtain a uniform bacterial load equivalent to $5 \times 10^4$ CFU/mL. Nutrient agar (NA)-containing Petri dishes were used to assess the zone of inhibition (ZOI) for each strain when incubated for 24 h at $37 \pm 1°C$. In brief, 25 mL of sterilized NA was allowed to cool to room temperature and then 1 mL of culture suspension was added to the NA and the mixture was thoroughly mixed. The mixture was poured into the Petri dish and, after solidifying, a sterilized stainless-steel borer was used to create a well (8 mm in diameter) in the same plate. Then, 100 μL of the test sample was dropped into the marked well, which was then covered and left for 4 h. The plates were inverted and incubated for 24 h. The study was conducted in triplicate for each strain to obtain means ± sd. The ZOI created around each well was measured using a scale and noted.

The antifungal activities of the compounds against *C. albicans* and *A. niger* were also estimated following the well-diffusion method [33]. Both strains were recultured in Czapek Dox medium (pH 6.8) and allowed to grow exponentially to $OD_{600} \sim 0.5$. *Aspergillus niger* was grown on solid Sabouraud Dextrose agar (SDA, pH 6.0). For this, aqueous-dispersed SDA medium was completely mixed and sterilized at 121°C for 15 min. Then, 25 mL of the medium was aseptically poured into the Petri dish and allowed to cool to room temperature. An aliquot (1 mL) of the culture was mixed with 25 mL of normalized medium followed by incubation for 48 h at 37 °C. After incubation, colonies were counted and serially diluted ($10^{-5}$ dilutions) to obtain $3 \times 10^5$ CFU/mL and $6 \times 10^5$ CFU/mL for *A. niger* and *C. albicans*, respectively. This fungal load was used for the study of ZOI. *Candida albicans* was incubated for 48 h at $35 \pm 1 °C$, whereas *A. niger* was incubated for 7 days. For the ZOI study, the same above-described method used for the bacteria was followed.

Assessment of Minimum Inhibitory Concentration (MIC)

To evaluate the MIC of the compounds, bacterial cultures were serially diluted ($10^{-5}$) to obtain a fixed CFU/mL for each strain [34]. Sterilized NA medium (25 mL), 0.1 mL of the culture, and the test sample (known concentration) were completely mixed at room temperature and allowed to solidify. Then, the Petri dish was incubated for 24 h at the temperatures specified in the previous section. After incubation, the absence of colonies on the plate indicated the MIC of the tested compound at the evaluated concentration. The MIC was defined as the minimum concentration required to completely inhibit bacterial colony formation. A similar method was adopted for the fungal strains. Isoniazid and rifampicin were used as positive controls for the antibacterial assay, whereas ketoconazole was used as a positive control for the fungal strains.

### 2.2.9. Measurement of Hemolysis

In this study, it was necessary to exclude the possibility that the synthesized imine compounds could induce hemolysis when administered transdermally or orally. To test this, an in vitro hemolysis assay was carried out using rat erythrocytes (red blood cells, RBCs). Suspensions of RBCs (4% *v/v*) were prepared in PBS solution (pH 7.4). The samples (F3 for **AM-3**, F5 for **AM-5**, F7 for **AM-7**, and F8 for **AM-8**) to be tested were diluted in 5% LAB aqueous solution, yielding final concentrations of 10 and 100 μg/mL. Saline and distilled water served as negative and positive controls, respectively. LAB solution, 5% DMSO, and PBS were used as placebo and dilution media. Briefly, 0.5 mL of the RBC suspension, 1 mL of the test sample, and 3.5 mL of PBS were gently mixed in a centrifugation tube and incubated for 4 h at $37 \pm 1°C$. After incubation, the samples were removed and centrifuged at 5000 rpm to separate RBCs and proteins (if lysed). The supernatants containing released hemoglobin were used to estimate the hemoglobin content by measuring the absorbance at 540 nm using a UV-Vis spectrophotometer. Percent hemolysis was calculated as per the following equation:

$$\text{Percent hemolysis} = [(A_S - A_N)/(A_P - A_N)] \times 100 \tag{1}$$

where $A_s$, $A_N$, and $A_P$ represent the absorbance of the test sample, negative control, and positive control, respectively. Positive control-induced hemolysis was considered as 100% [35].

## 3. Results and Discussion

### 3.1. Nuclear Magnetic Resonance Spectroscopy

The imines and imine-based analogs were synthesized via the routes shown in Figure 2. The introduction of pharmacophores in formyl heterocycles has been shown to result in biologically active analogs [36]. A substantial number of systematic studies have focused on employing C-N bond formation under strongly acidic conditions for the synthesis of various heterocyclic molecules. However, the reaction requires prolonged heating and is tedious. In this study, one-step route was used to synthesize imines through the reaction of 3-formylthiophene with substituted *N*-nucleophile, and expected for potential antimicrobial activities. The structures of **AM-3**, **AM-5**, **AM-7,** and **AM-8** were obtained by $^1$H NMR spectra with respect to the exocyclic C-H proton which appears as a singlet, which confirmed the desired products. The synthesized compounds were well differentiated by their spectral fingerprint (Figures 3 and 4). FTIR spectroscopy further supported the proposed structures [36].

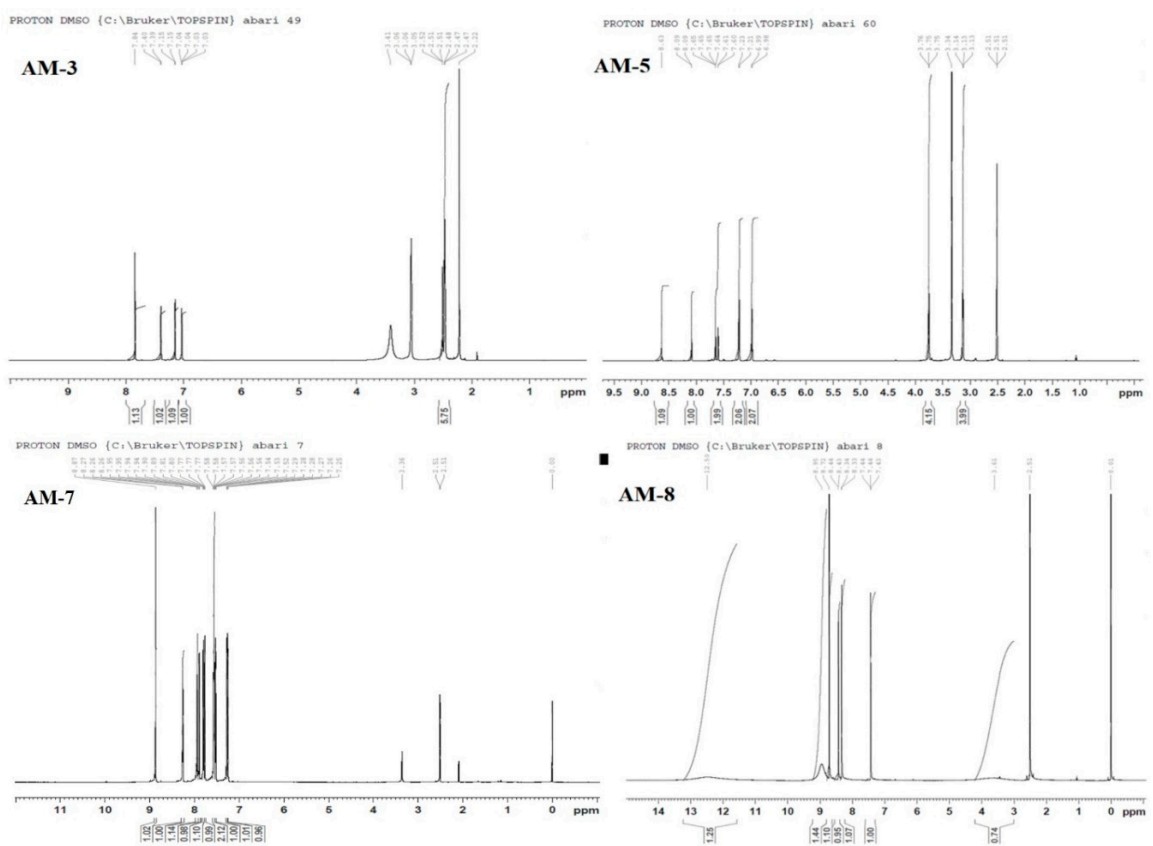

**Figure 4.** $^1$H NMR spectra of **AM-3**, **AM-5**, **AM-7**, and **AM-8**.

**AM-3**: 4-methyl-N-(thiophen-2-ylmethylene)piperazin-1-amine
$^1$HNMR (700.124 MHz, DMSO-$d_6$): δ = 2.2 (s, 3H, *N*-CH$_3$), 2.47 (t, 4H, *j* = 7.0 Hz, 2 × CH$_2$), 3.05 (t, 4H, *j* = 7.0 Hz, 2 × CH$_2$), 7.04 (dd, $j_{1,4}$ = 7.0 Hz, 1H, thiophene), 7.15 (d, 1H, *j* = 3.5 Hz, thiophene), 7.39 (d, 1H, *j* = 7.0 Hz, thiophene), 7.84 (s, 1H, CH).
**AM-5:** 4-morpholino-N-(thiophen-2-ylmethylene)aniline

$^1$HNMR (700.124 MHz, DMSO-$d_6$): δ = 3.1 (t, 4H, *j* = 14 Hz, 2 × CH$_2$), 3.75 (t, 4H, *j* = 7.0 Hz, 2 × CH$_2$), 6.99 (d, 2H, *j* = 7.0 Hz, arom), 7.21 (d, 2H, *j* = 14 Hz, arom), 7.60 (d, 1H, *j* = 7.0 Hz, thiophene), 7.65 (dd, 1H, $j_{1,4}$ = 7.0 Hz, thiophene), 8.09 (s, 1H, thiophene), 8.63 (s, 1H, CH).

**AM-7**: N-(thiophen-2-ylmethylene)naphthalen-1-amine

$^1$HNMR (700.124 MHz, DMSO-$d_6$): δ = 7.25 (d, 1H, *j* = 7.0 Hz, arom), 7.28 (dd, 1H, $j_{1,4}$ = 14 Hz, arom), 7.52 (t, 1H, *j* = 7.0 Hz, arom), 7.56 (m, 2H, arom), 7.77 (t, 1H, *j* = 3.5 Hz, arom) 7.80 (d, 1H, *j* = 8.4 Hz, arom), 7.89 (d, 1H, *j* = 7.0 Hz, thiophene), 7.94 (m, 1H, thiophene), 8.26 (dd, 1H, $j_{1,4}$ = 7.0 Hz, thiophene), 8.87 (s, 1H, CH). The $^{13}$C NMR report for **AM-7** is provided in Supplementary Figure S1.

**AM-8:** 2-([thiophen-2-ylmethylene]amino)pyrimidine-4,6-diol

$^1$HNMR (700.124 MHz, DMSO-$d_6$): δ = 7.44 (t, 1H, *j* = 7.0 Hz, arom), 8.34 (d, 1H, *j* = 7.0 Hz, thiophene), 8.44 (d, 1H, *j* = 7.0 Hz, thiophene), 8.72 (s, 1H, CH), 12.50 (br s, 1H, OH). The $^{13}$C NMR report for **AM-8** is provided in Supplementary Figure S2.

### 3.2. Fourier Transform Infrared Spectroscopy

Each synthesized product showed distinguishing peaks (fingerprints) in the IR spectrum (Figure 5). Compound **AM-3** exhibited characteristic peaks at 2790.07, 1443.55 (C-C stretching vibration of thiophene), 1357.21 (C-N and N-N stretching vibration), 1278.82 (in-plane C-H stretching vibration of thiophene), 1145.5 (C-H bending vibration), 991.33, 889.64, and 799.05 cm$^{-1}$, wherein the C-H stretching vibration of the thiophene ring was in two modes, namely, in-plane and out-of-plane vibration stretching at 1257.21 cm$^{-1}$ and 991.33–799.05 cm$^{-1}$, respectively. The stretching vibration at 1145.5 cm$^{-1}$ was characteristic of the C-S bond of thiophene in **AM-3** [37]. The piperazine ring was identified via two stretching vibrations due to asymmetrical C-H stretching and a symmetrical C-H stretching vibration at 2790.09 cm$^{-1}$ and 2980 cm$^{-1}$, respectively. Notably, the stretching vibration identified at 1357.21 cm$^{-1}$ was due to the formation of an N-N bond in **AM-3** and the N-N stretching mode, which showed a slight redshift from 1374.0 cm$^{-1}$ [38].

All the synthesized compounds (**AM-3**, **AM-5**, **AM-7**, and **AM-8)** possessed thiophene as a moiety and therefore the C-H (in-plane and out-of-plane stretching), C-C, and C-S (1145.0 or 1072.0 cm$^{-1}$) stretching vibrations showed a slight shift in wavenumber (C-S stretching) and appeared at 1109.59 cm$^{-1}$ (**AM-5**), 1030.82 cm$^{-1}$ (**AM-7**), and 1068.87 cm$^{-1}$ (**AM-8**). Compounds **AM-5** and **AM-7** could easily be identified through the characteristic C-N stretching in the aromatic amine, as shown in Figure 3. Moreover, **AM-5** possessed a morpholine ring and was identified via two transmission signals, one at 1600.58 cm$^{-1}$ (C-N stretching between benzene and the morpholine ring) and the other at 1490.30 cm$^{-1}$ (C-O stretching of the morpholine ring). The vibrational stretching observed at 1229.99 cm$^{-1}$ revealed a redshift from 1239 cm$^{-1}$ for C-O-C stretching [39]. Similarly, compound **AM-7** was identified through the stretching vibration of its aromatic amine (C-N) in the range of 1382–1266 cm$^{-1}$ (theoretical wavenumber) which appeared at 1381.65 and 1207.84 cm$^{-1}$ (observed wavenumber). Finally, compound **AM-8** is extremely lipophilic and contains both thiophene and a pyrimidine ring (Figure 3). **AM-8** displayed C=N stretching vibration at 1549.80 cm$^{-1}$. An additional common stretching vibration that appeared at 1360.78 cm$^{-1}$ for **AM-5**, 1381.65 cm$^{-1}$ for **AM-7**, and 1393.54 cm$^{-1}$ for **AM-8** was due to C-N-C in-plane bending of the imine formed. Similarly, the stretching vibration at 714.54 cm$^{-1}$ (**AM-5**), 773.99 cm$^{-1}$ (**AM-7**), and 744.0 cm$^{-1}$ (**AM-8**) represented the C-N-C out-of-plane bending of the imine formed. These findings are in agreement with those reported for imine (1369 cm$^{-1}$ in-plane, 747 cm$^{-1}$ out-of-plane) [40].

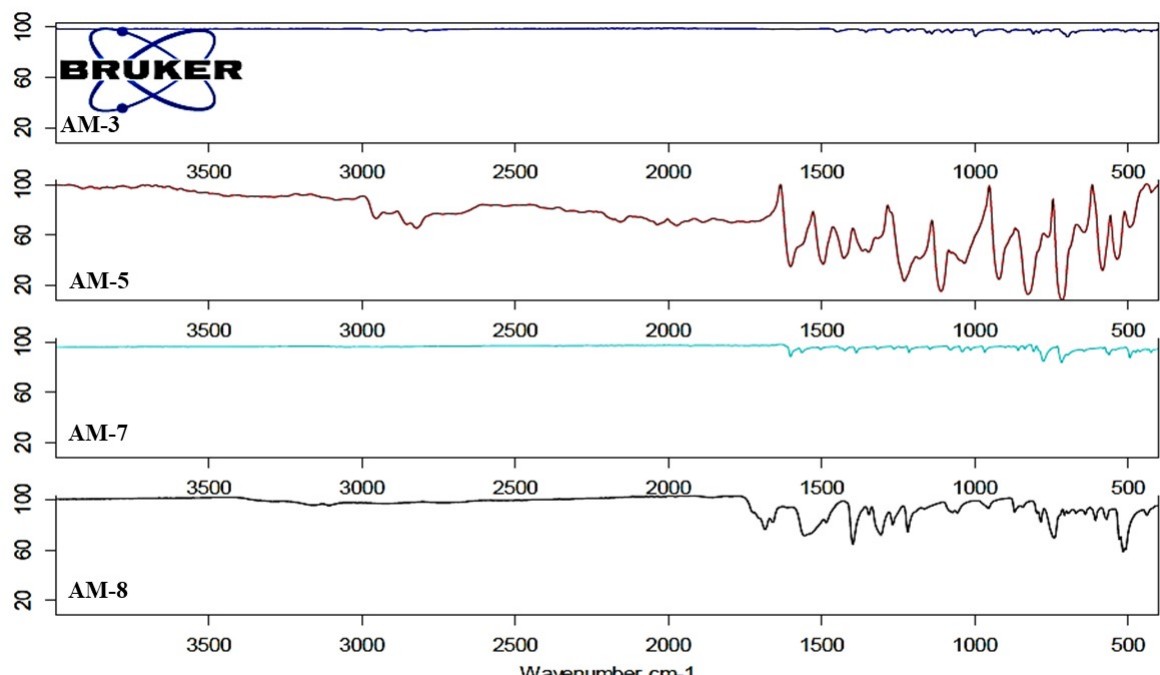

**Figure 5.** FTIR analysis for **AM-3**, **AM-5**, **AM-7**, and **AM-8**.

### 3.3. Differential Scanning Calorimetry

All the compounds were analyzed for thermal behavior, and the thermograms are depicted in Figure 6. Compounds **AM-3**, **AM-5**, **AM-7**, and **AM-8** exhibited characteristic endothermic peaks at 70.42 °C, 197.82 °C, 80.53 °C, and 279.97 °C, respectively. These values corresponded to their melting points (fusion temperature). The absence of other peaks was indicative of the purity of the compounds and their thermal stability under the explored temperatures. The observed crystalline nature of the imine compounds may explain their insolubility in aqueous medium.

### 3.4. Powder X-ray Diffraction

To confirm the solid-state characteristics of the synthesized compounds, we performed PXRD (Figure 7). Compound **AM-3** showed characteristic low-intensity peaks at the 2θ values of 9.4°, 18.0°, and 22.5°. Similarly, **AM-7** displayed multiple characteristic peaks at the 2θ values of 17.3°, 18.4°, 20.8°, and 24.1°. Notably, the peaks for **AM-5** and **AM-8** were more intense than those for **AM-3** and **AM-7,** as shown in Figure 7. **AM-5** displayed characteristic 2θ values of 18.9°, 21.4°, 24.2°, 28.1°, and 41.7°, whereas those for **AM-8** were 11.9°, 15.7°, 20.1°, 25.4°, 27.9°, and 31.2°. These characteristics confirmed the crystalline nature of the four synthesized compounds. Additionally, these findings were also in agreement with the DSC results and with the water-insoluble nature of the compounds.

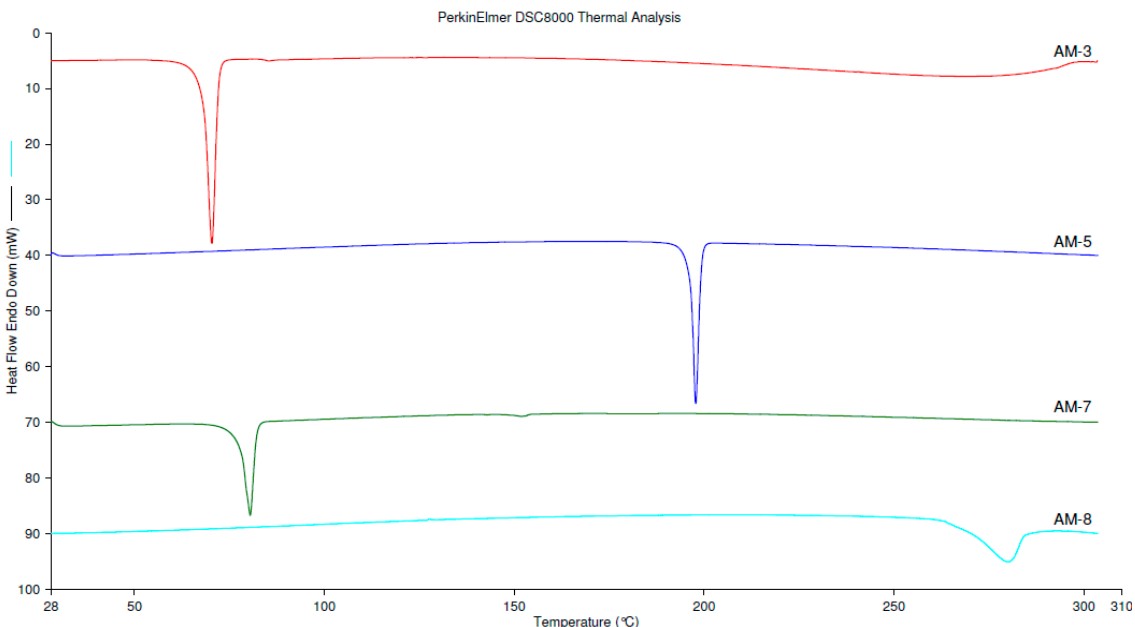

**Figure 6.** Differential scanning calorimetry thermograms for **AM-3**, **AM-5**, **AM-7**, and **AM-8**.

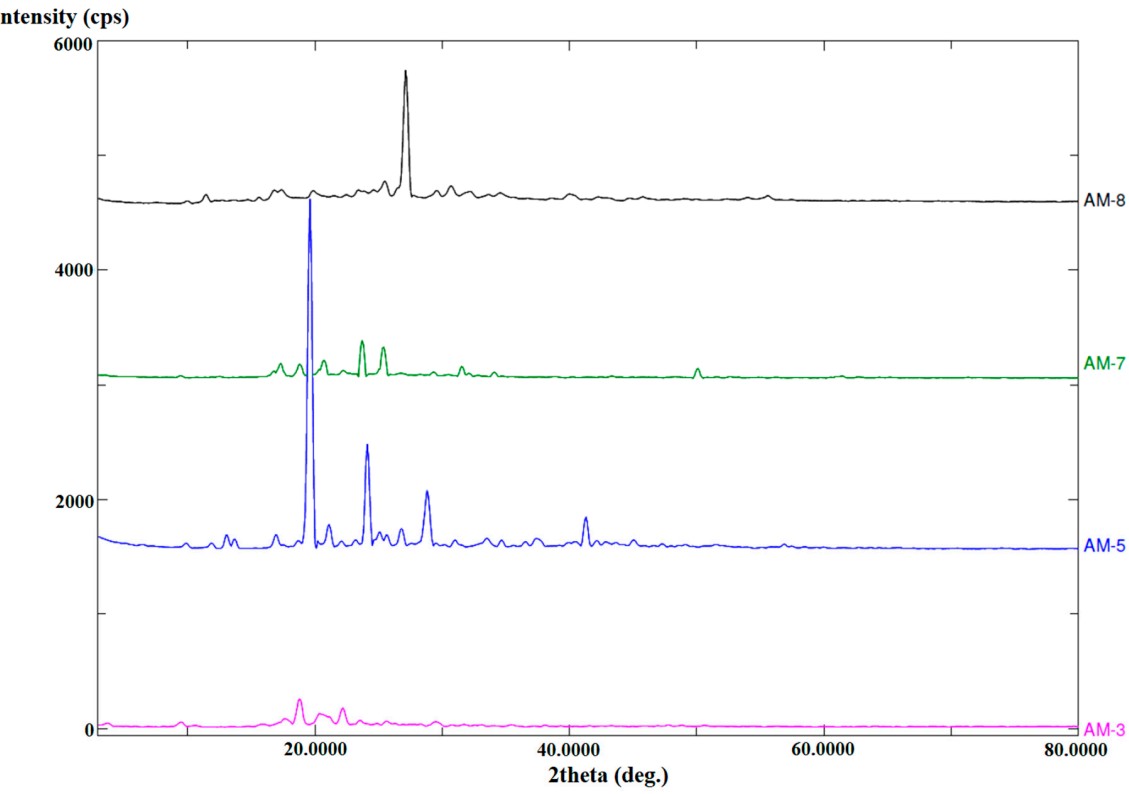

**Figure 7.** Powder X-ray diffraction (PXRD) patterns for **AM-3**, **AM-5**, **AM-7**, and **AM-8**.

### 3.5. Assessment of Morphology and Composition using EDX-SEM

SEM is the most advanced technology currently available for high-magnification microscopic evaluation, and we used this technique to assess the morphology and composition of the solid powder form of the four compounds. As can clearly be seen in Figure 8, all the synthesized imine compounds were highly crystalline and solid. The solid crystal form of compounds **AM-5** and **AM-8** was more acicular (Figure 8B,D, respectively). The crystals consisted of broken pieces with smooth surface

morphology. In contrast, **AM-3** and **AM-7** exhibited soft surfaces with a stone-like morphology (Figure 8A,C, respectively). A similar morphology has been reported with synthesized compounds containing a Schiff base [41]. Furthermore, EDX SEM analysis confirmed the elemental content of the synthesized compounds (Figure 9A–D). The results confirmed the presence of C, N, H, and S in the molecular structure. The empirical formula for **AM-3** is $C_{10}H_{15}N_3S$, representing a theoretical percent composition of 34.48%, 10.34%, 51.72%, and 3.4% for C, N, H, and S, respectively. These values were found to be closest as obtained from EDX analysis which supported the synthesized molecule for the proposed structure of AM-3. However, the EDX SEM results showed percent composition values of 36.93%, 10.84%, 47.82%, and 5.42% for C, N, H, and S, respectively, which were very close to the theoretical values (Figure 8A). **AM-7** ($C_{15}H_{11}NS$) has a theoretical percent composition of 88.23, 5.88, and 5.88% for C, N, and S, respectively, whereas the respective values obtained from the EDX SEM analysis were 78.74%, 16.44%, and 4.87%. The empirical formulas for **AM-5** and **AM-8** are $C_{15}H_{16}N_2OS$ and $C_9H_7N_3O_2S$, respectively. The theoretical percent composition of **AM-5** was found to be 53.57%, 3.57%, 39.28%, and 3.57% for C, N, H, and S, respectively, whereas the corresponding values obtained from the EDX SEM analysis were 68.23%, 6.16%, 21.39%, and 4.22%. For **AM-8**, the theoretical percent composition was 60.0%, 20.0%, 13.33%, and 6.66% for C, N, O, and S, respectively, whereas the respective values were 62.76%, 18.71%, 12.79%, and 5.75% as obtained by EDX SEM analysis (Table 1). These findings confirmed the empirical formulas and elemental purity of the samples. In brief, these compounds were found to be crystalline and solid based on the PXRD and DSC analyses.

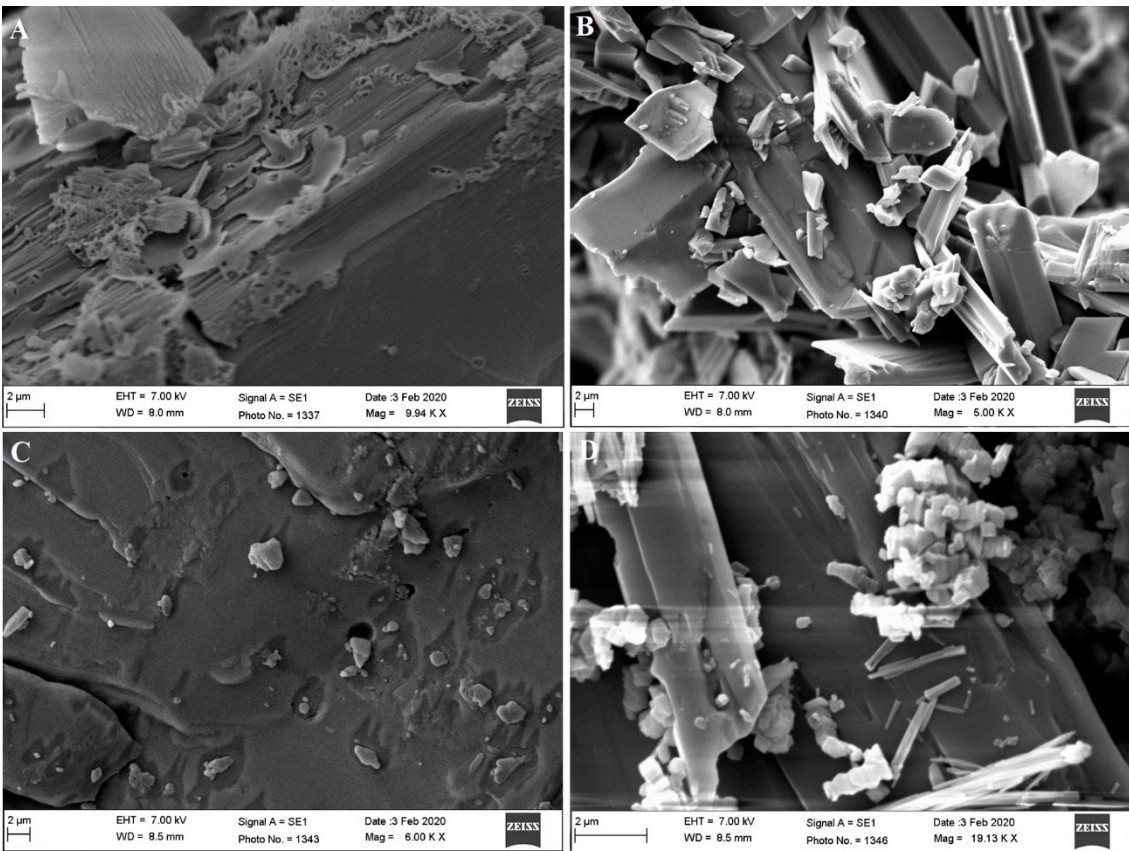

**Figure 8.** Micrographs of the surface morphology showing (**A**) **AM-3**, (**B**) **AM-5**, (**C**) **AM-7**, and (**D**) **AM-8**.

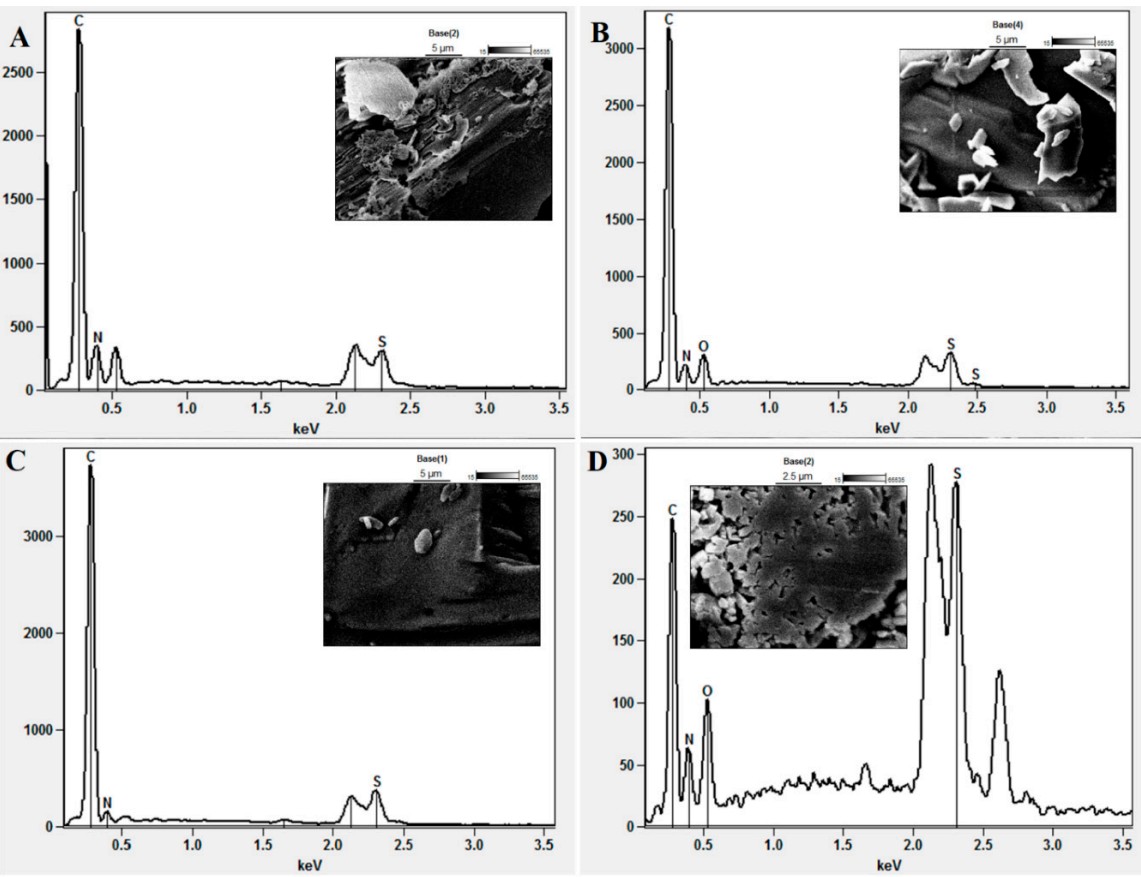

**Figure 9.** Energy-dispersive X-ray spectroscopy (EDX) coupled with scanning electron microscopy (SEM) (EDX SEM) showing (**A**) **AM-3**, (**B**) **AM-5**, (**C**) **AM-7,** and (**D**) **AM-8**.

**Table 1.** SEM EDX report for the percent composition of the elements present in the compounds.

| Scanning specification | EDX summary report on scanned image | | | |
|---|---|---|---|---|
| | AM-3 | AM-5 | AM-7 | AM-8 |
| Image Resolution | 512 × 384 | 512 × 384 | 512 × 384 | 512 × 384 |
| Image Pixel Size (μm) | 0.06 | 0.06 | 0.06 | 0.06 |
| Map Resolution | 256 × 192 | 256 × 192 | 256 × 192 | 256 × 192 |
| Map Pixel Size (μm) | 0.11 | 0.11 | 0.11 | 0.11 |
| Acc. Voltage (kV) | 7 | 7 | 7 | 7 |
| Magnification | 10,000 | 10,000 | 10,000 | 10,000 |
| Take-off angle | 33.4° | 35.0° | 35.0° | 35.0° |
| Element scanned | Composition analysis report (%) | | | |
| | AM-3 | AM-5 | AM-7 | AM-8 |
| Carbon | 53.77 | 68.23 | 78.70 | 42.26 |
| Nitrogen | 40.82 | 21.39 | 16.44 | 18.71 |
| Oxygen | – | 6.16 | – | 9.28 |
| Sulphur | 5.51 | 4.22 | 4.87 | 29.75 |

*3.6. Solubility Study in Various Solvents and Surfactants*

The results of the solubility assessment are depicted in Figure 10. All the compounds were found to be insoluble in water, which was due to the crystalline and hydrophobic nature of the imine-containing (Schiff base) compounds. **AM-3, AM-5,** and **AM-7** showed maximum solubility in chloroform (32.2 ± 1.61, 44.5 ± 2.2, and 85.3 ± 4.2 mg/mL, respectively). Compound **AM-8** displayed maximum

solubility in Miglyol-812 (7.9 mg/mL) and Cremophor-EL (8.0 mg/mL), which may have resulted from the lipophilic–lipophilic (solute–solvent) interaction between the compound and Miglyol-812 or Cremophor-EL. Insolubility in water, Tween 80 (10%), and ethanol may be correlated with the inability of **AM-8** (two hydroxyl groups) to form hydrogen bonds with water molecules. Similarly, **AM-3**, **AM-5,** and **AM-7** lack functional groups that can form hydrogen bonds with water, and hence are also insoluble in water and Tween 80. The solubility of a compound depends on various factors, such as the molar mass of the solute and solvent; solute–solute, solvent–solvent, and solute–solvent interactions; temperature; and the physicochemical nature of the solute in the investigated solvent [41].

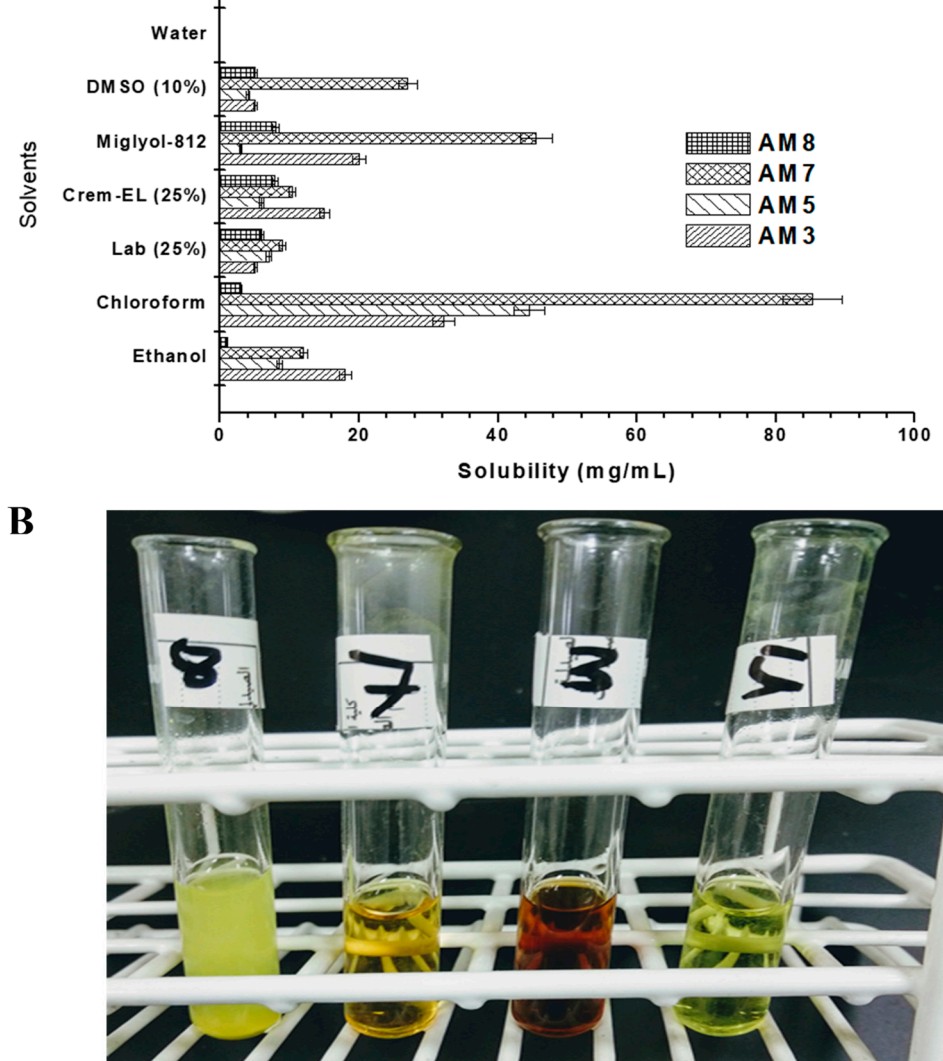

**Figure 10.** Solubility assessment in various solvents and surfactants at 30 ± 1 °C. (**A**) Drug solubility in different excipients. (**B**) The solubility of **AM-3**, **AM-5**, and **AM-7**, in chloroform at 30 ± 1 °C.

### 3.7. In Vitro Antimicrobial Activities

#### 3.7.1. Zone of Inhibition (ZOI)

To determine whether the synthesized imines possessed antimicrobial activity, we evaluated the ZOI generated by the four compounds for eight microbial species, as shown in Table 2. None of the

compounds exhibited any activity against *S. aureus*, whereas **AM-3** showed potential activity against the other five bacterial species and the two fungal species (Table 2). The sizes of the ZOIs for **AM-3** were 16.0 ± 0.81, 15.6 ± 0.78, 18.0 ± 0.91, 16.0 ± 0.82, 14.6 ± 0.73, 15.3 ± 0.76 m, and 17.3 ± 0.86 mm for *B. subtilis*, *E. coli*, *A. baumannii*, *M. smegmatis*, *E. faecalis*, *C. albicans*, and *A. niger*, respectively. For **AM-5,** the ZOIs were 9.3 ± 0.46, 7.3 ± 0.36, and 9.7 ± 0.48 mm against *B. subtilis*, *E. coli*, and *A. baumannii*, respectively, and 9.7 ± 0.48 and 9.3 ± 0.47 mm against *C. albicans* and *A. niger*, respectively. The sizes of the ZOIs for compound **AM-7** were 12.0 ± 0.65, 16.3 ± 0.81, 16.0 ± 0.83, 14.3 ± 0.71, 14.3 ± 0.73, and 15.7 ± 0.79 mm against *B. subtilis*, *E. coli*, *A. baumannii*, *M. smegmatis*, *C. albicans* and *A. niger*, respectively. Compound **AM-8** exhibited inhibitory activity against *B. subtilis*, *E. coli*, and *A. baumannii* only (Table 3), with ZOI values of 5.3 ± 0.26, 8.6 ± 0.43, and 7.3 ± 0.36 mm, respectively. Among all the compounds, **AM-7** exhibited the most wide-ranging antibacterial and antifungal activity, which may have been due to its higher solubility in DMSO and LAB, the solvents that were used to solubilize the compounds. The absence of the generation of a ZOI by the solvents in isolation confirmed that the generated ZOIs were due to the compound only. **AM-8** also showed antifungal activity (Table 2). Notably, compounds **AM-3** and **AM-7** exhibited inhibitory activity only against *M. smegmatis*, which may have been due to the naphthalene-1-amine and methyl piperazine moieties present in this molecule [42]. Paengsri et al. reported that a 1,4-benzoquinone derivative exerted potential antimycobacterial effects on *M. tuberculosis* strain H$_{37}$Rv [42,43]. Compound **AM-5** (4-morpholino-N-[thiophen-2-yl methylene]aniline) showed significant inhibitory activities against *C. albicans* and *A. niger*. The antifungal potential of this novel 4-morpholino derivative may be attributable to the presence of the basic 4-morpholino-N-aniline moiety, similar to that reported for a Schiff base-containing synthesized 4-morpholino derivative that showed an MIC~30 µg/mL [44]. Thus, the **AM-5** and **AM-8** are the only sensitive against Gram-negative bacteria as shown in Table 2. Various synthesized pyrimidine derivatives are reported to possess antifungal, antibacterial, and antiviral properties [45], suggesting that the antibacterial effect of **AM-8** may be due to its pyrimidine-1,4-diol moiety.

### 3.7.2. Minimum Inhibitory Concentration

The MIC values are presented in Table 3. *Escherichia coli*, *M. smegmatis*, and *A. niger* showed the greatest sensitivity to compound **AM-3**, with an MIC value of 31.25 µg/mL. Compound **AM-5** showed potential antibacterial activity against *E. coli* and *C. albicans* at an MIC of 25 µg/mL, whereas *M. smegmatis* and *E. faecalis* were not affected by AM-5 at the concentration tested, which may have been due to the low solubility of the compound and its accumulation in the *M. smegmatis* cell wall to a toxic level. Compound **AM-7** exhibited the most pronounced antibacterial and antifungal activities of all the compounds at the lowest concentration tested (Table 3), which may be associated with its greater solubility. The MIC values of these compounds suggested that **AM-3**, **AM-5**, **AM-7,** and **AM-8** may be effective against Gram-negative bacteria and fungal strains, but not against Gram-positive bacteria. The MIC results obtained with the positive controls (isoniazid, rifampicin, and ketoconazole; Table 3) were in agreement with previously reported findings [46–48].

**Table 2.** In vitro antimicrobial assay for **AM-3**, **AM-5**, **AM-7,** and **AM-8** against various microbial species.

| Compounds | Zone of Inhibition (mm) * | | | | | | | |
|---|---|---|---|---|---|---|---|---|
| | *S. aureus* | *B. subtilis* | *E. coli* | *A. baumannii* | *M. smegmatis* | *E. faecalis* | *C. albicans* | *A. niger* |
| **AM-3** | – | 16.0 (0.81) | 15.6 (0.78) | 18.0 (0.91) | 16.0 (0.82) | 14.6 (0.73) | 15.3 (0.76) | 17.3 (0.86) |
| **AM-5** | – | 9.3 (0.46) | 7.3 (0.36) | 9.7 (0.48) | – | – | 9.7 (0.48) | 9.3 (0.47) |
| **AM-7** | – | 12.0 (0.65) | 16.3 (0.81) | 16.0 (0.83) | 14.3 (0.71) | – | 14.7 (0.73) | 15.7 (0.79) |
| **AM-8** | – | 5.3 (0.26) | 8.6 (0.43) | 7.3 (0.36) | – | – | 7.3 (0.35) | 7.7 (0.39) |
| **DMSO (5%)** | – | – | – | – | – | – | – | – |
| **LAB (5%)** | – | – | – | – | – | – | – | – |

*S. aureus*: *Staphylococcus aureus*; *B. subtilis*: *Bacillus subtilis*; *E. coli*: *Escherichia coli*; *A. baumannii*: *Acinetobacter baumannii*; *M. smegmatis*: *Mycobacterium smegmatis*; *E. faecalis*: *Enterococcus faecalis*; *C. albicans*: albicans; *Aspergillus Niger*: *Aspergillus Niger*; LAB: labrasol. * Values in parentheses are the standard deviation of the mean (*n* = 3).

**Table 3.** The calculated minimum inhibitory concentrations (MICs) for **AM-3**, **AM-5**, **AM-7,** and **AM-8** against various microbial species.

| Compound | Minimum Inhibitory Concentration (µg/mL) | | | | | | | |
|---|---|---|---|---|---|---|---|---|
| | *S. aureus* | *B. subtilis* | *E. coli* | *A. baumannii* | *M. smegmatis* | *E. faecalis* | *C. albicans* | *A. niger* |
| **AM-3** | – | 62.5 | 31.25 | 125.0 | 31.25 | 125.0 | 62.5 | 31.25 |
| **AM-5** | – | 100.0 | 25.0 | 200.0 | – | – | 25.0 | 100.0 |
| **AM-7** | – | 31.25 | 31.25 | 125.0 | 15.62 | – | 62.5 | 15.62 |
| **AM-8** | – | 100.0 | 50.0 | 12.5 | – | – | 50.0 | 50.0 |
| **\* Ketoconazole** | – | – | – | – | – | – | 0.5 | 0.25 |
| **\* Isoniazid** | NS | NS | NS | NS | 0.125 | NS | NS | NS |
| **\* Rifampicin** | 0.25 | 0.15 | 16 | 32 | 0.64 | 8.0 | NS | NS |
| **DMSO (5%)** | – | – | – | – | – | – | – | – |
| **LAB (5%)** | – | – | – | – | – | – | – | – |

*S. aureus*: *Staphylococcus aureus*; *B. subtilis*: *Bacillus subtilis*; *E. coli*: *Escherichia coli*; *A. baumannii*: *Acinetobacter baumannii*; *M. smegmatis*: *Mycobacterium smegmatis*; *E. faecalis*: *Enterococcus faecalis*; *C. albicans*: *Candida albicans*; *Aspergillus Niger*: *Aspergillus Niger*; LAB: labrasol; NS = not studied; DMSO = dimethyl sulfoxide. Note: * Positive control.

### 3.8. In Vitro Hemolysis Study

In the present study, we aimed to synthesize hemocompatible and safe novel imine compounds possessing substantial antibacterial and antifungal properties. Therefore, it was necessary to ensure their hemocompatibility with erythrocytes at the dose range of 10–100 µg/mL used in the in vitro antimicrobial assay (Figure 11). The results showed that positive and negative controls caused 100% and 11.74% hemolysis, respectively. PBS, used as a dilution medium, showed results similar to those obtained with the negative control. The solvents (DMSO and LAB) were found to be safe at the 5% *v/v* concentration used to solubilize the compounds. **AM-3, AM-5, AM-7,** and **AM-8** caused hemolysis at levels (≤12.8%) similar to those of the negative control at both the 10 and 100 µg/mL concentrations. These findings suggested that the concentrations that were lethal for the bacterial and fungal strains used in this study are not toxic for erythrocytes and can, therefore, be used for oral and systemic delivery at a suitable dose after establishing further acute and chronic toxicity data in preclinical studies.

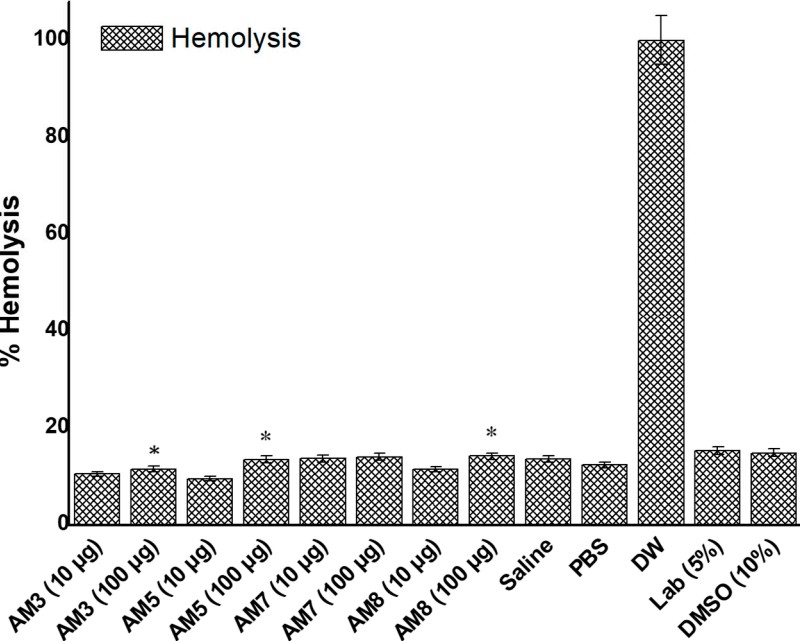

**Figure 11.** In vitro hemolysis evaluation of two concentrations of the synthesized compounds using rat erythrocytes (* $p > 0.05$ compared with the negative control). DW represents distilled water, a positive control for 100% hemolysis; PBS represents the dilution medium.

Imine compounds have been synthesized for various therapeutic applications, such as phthalimide imine derivatives exhibiting anticancer properties and poly(imine) dendrimers functionalized with an organometallic moiety synthesized as drugs against cisplatin-resistant cancer cells [49,50]. These were evaluated for cytotoxicity and anticancer potential in vitro using a human colon cancer cell line. The authors reported selective cancer inhibition (breast and colon) without toxicity to normal tissue [49,50]. Consequently, we plan to evaluate the compounds synthesized in this study (**AM-3, AM-5, AM-7,** and **AM-8**) for their cytotoxic potential against the human colon cell line HT-29, thereby providing further evidence for their safety for oral delivery.

### 4. Conclusions

Bacterial and fungal infections cover a wide range of systemic and local diseases. The drugs that are currently available to treat these diseases face various clinical challenges for therapeutic application, and novel synthesized Schiff bases may be suitable alternatives to conventional antibacterial and antifungal drugs. The evaluation parameters of the physicochemical characterization suggested that these compounds are hydrophobic and crystalline solids, and thereby require further assessment for

improved water solubility using suitable cosolvents. Nevertheless, our findings that these compounds exhibited antifungal and antibacterial activities indicated that they may be suitable for controlling bacterial and fungal infections after topical or oral administration. Furthermore, all the compounds were found to be hemocompatible at the evaluated concentrations. However, a long-term acute toxicity study using a suitable animal model must be established to corroborate the safety aspects.

**Supplementary Materials:** The following are available online at http://www.mdpi.com/2227-9717/8/11/1476/s1, Figure S1: 13C NMR spectra's of AM-7, Figure S2: 13C NMR spectra's of AM-8.

**Author Contributions:** M.A.A.: conceptualization, software and funding, A.H.: conceptualization, software, writing—original draft preparation, S.A.: data curation and reviewing, S.S.I.: software, validation, A.B.: writing and analysis, A.A.: data curation and reviewing. All authors have read and agreed to the published version of the manuscript.

**Funding:** All are thankful to the Deputyship for Research & Innovation "Ministry of Education" in Saudi Arabia for funding this research work through the project number IFKSURG-1441-010.

**Acknowledgments:** The authors extend their sincere appreciation to the Deputyship for Research & Innovation "Ministry of Education" in Saudi Arabia for funding this research work through the project number IFKSURG-1441-010.

**Conflicts of Interest:** The authors declare no conflict of interest.

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
