# Peer review of "Novel Hemocompatible Imine Compounds as Alternatives for Antimicrobial Therapy in Pharmaceutical Application"

_processes, doi:10.3390/pr8111476_

Round 1

Reviewer 1 Report

In this study the authors report the synthesis of neoteric imines and present their characterization by NMR, FTIR, DSC and PXRD. Authors also present the in vitro studies of the effect of these imines as antimicrobial antibacterial and antifungal agents to control local and systemic infections. The aim of the work is highly ambitious, and the results are very promising.

Characterization of the imine derivatives through diverse techniques are suitable and clear. The antimicrobial studies are adequate and well explained. Nevertheless, due to huge challenge that is control systemic infections, it is important to emphasize that this survey is a preliminary study and some discussion about the points below must be included:

- No attempt was made to determine if the imine derivatives damage plasma clotting factors. Do the authors have any idea if these class of compounds have some effect in these factors? What the literature says about this?

- Since the idea is to delivery this compound orally, I think that is reasonably to include in this study the cytotoxicity of the imine compounds in human cell lines or at least discuss this point based in literature information about the cytotoxicity of these class of organic compounds.

Author Response

Response letter                                                                                                 Date: 25-10-2020

Manuscript ID: processes-974914

Title: Novel hemocompatible imine drugs as new alternative for antimicrobial therapy in pharmaceutical application

Reviewer 1: In this study the authors report the synthesis of neoteric imines and present their characterization by NMR, FTIR, DSC and PXRD. Authors also present the in vitro studies of the effect of these imines as antimicrobial antibacterial and antifungal agents to control local and systemic infections. The aim of the work is highly ambitious, and the results are very promising.

Characterization of the imine derivatives through diverse techniques are suitable and clear. The antimicrobial studies are adequate and well explained. Nevertheless, due to huge challenge that is control systemic infections, it is important to emphasize that this survey is a preliminary study and some discussion about the points below must be included:

Comments and Suggestions for Authors

Comment 1: - No attempt was made to determine if the imine derivatives damage plasma clotting factors. Do the authors have any idea if these class of compounds have some effect in these factors? What the literature says about this?  

Response 1: Dear sir, this is our preliminary study of a major project. At this stage, we conducted hemolysis study using erythrocytes (an in vitro study) which gives a minimum safety concern about the toxicity. I could not find any literature claiming about the detrimental effect of imine on plasma clotting factors. We planned for short and long term toxicity studies using animal model for evaluation of safety of the compound. In that, I will take this factor too.   

Comment 2: - Since the idea is to delivery this compound orally, I think that is reasonably to include in this study the cytotoxicity of the imine compounds in human cell lines or at least discuss this point based in literature information about the cytotoxicity of these class of organic compounds.

Response 2: Yes. This is an important study of these novel compounds. No such reports have been published on these synthesized imines. Therefore, colorectal cell-lines study is going on to determine IC50 of each compounds which would be published along with toxicity studies in our upcoming publication. Moreover, I added a literature support and a new paragraph on human colorectal cell lines study for imine based compounds. All of the changes have been addressed with red coloured text in revised manuscript. Thank you for your valuable suggestion regarding plasma clotting factor associated with imine compounds.

……………………………..Thank you for appreciable and valuable comments……………………….

Reviewer 2 Report

Introduction: Please elaborate on the introduction section. Your objectives and related background are unclear.  

Figure 10 B: Please can you elaborate on the figure caption.

Section 3.7.1 & Table 2: Images of ZOI for each compound would be more convincing.

Table 3 caption: Recommending using the word ‘calculated’ instead of ‘estimated’.  

Figure 11: Please can you calculate the significant difference between bars. Because AM3, 5 & 8 the % hemolysis increased with an increase in concentration.

Conclusion: Precise conclusion of obtained results followed by possible applications is expected in this section. I recommend rewriting this.  

Author Response

Response letter                                                                                                 Date: 25-10-2020

Manuscript ID: processes-974914

Title: Novel hemocompatible imine drugs as new alternative for antimicrobial therapy in pharmaceutical application

Reviewer 2: Comments and Suggestions for Authors

Comment 1: Introduction: Please elaborate on the introduction section. Your objectives and related background are unclear.  

Response 1: I have elaborated the introduction section as per suggestion. These changes were highlighted with red text in revised paper. Now, the section is clear and elaborated.

Comment 2: Figure 10 B: Please can you elaborate on the figure caption.

Response 2: I have revised the caption of figure 10B as per suggestion. The changes were highlighted with red text.

Comment 3: Section 3.7.1 & Table 2: Images of ZOI for each compound would be more convincing. 

Response 3: The results were presented in the table and therefore, it was not required for images for duplicity in results. I was better to present in table with mean values. 

Comment 4: Table 3 caption: Recommending using the word ‘calculated’ instead of ‘estimated’.

Response 4: I have replaced the word “estimated” to “calculated”. Thank you for suggestion.

Comment 5: Figure 11: Please can you calculate the significant difference between bars. Because AM3, 5 & 8 the % hemolysis increased with an increase in concentration. 

Response 5: Thank you for noticing a missing point. I have recalculated % hemolysis difference at p = 5%. However, there was no significant difference as compared to negative control group. Therefore, I have replaced the figure 11 with marked “* symbol” sign for AM-3, AM-5 and AM-8 followed by caption. The changes were addressed in red highlighted text.

Comment 6: Conclusion: Precise conclusion of obtained results followed by possible applications is expected in this section. I recommend rewriting this. 

Response 6: This section was re-written as per suggestion to make it clear and concise.

 ……………………………..Thank you for appreciable and valuable comments……………………….

Reviewer 3 Report

Please consider the following when revising the manuscript:

  1. Add a positive control in the in vitro antimicrobial assay;
  2. Describe "DW" in 2.2.9 and Figure 11;
  3. Correct "Parkinson's" in Line 67;
  4. Rewrite the legend for Figure 2;
  5. Move the abbreviations after the chemical names in the legend of Figure 3;
  6. Describe how the compounds were purified in Line 113;
  7. Change "drugs" to "compounds" or "agents" in the title (Line 2);
  8. Change the "0.0" to "-" in Table 3.

Author Response

Response letter                                                                                                 Date: 25-10-2020

Manuscript ID: processes-974914

Title: Novel hemocompatible imine drugs as new alternative for antimicrobial therapy in pharmaceutical application

Reviewer 3: Comments and Suggestions for Authors. Please consider the following when revising the manuscript:

Comment 1: Add a positive control in the in vitro antimicrobial assay?

Response 1: I have added the results of our positive control groups and revised the experimental and results and discussion part accordingly. These changes have been highlighted with red text. The table 3 has been revised too.

Comment 2: Describe "DW" in 2.2.9 and Figure 11?

Response 2: I have defined “DW” and changed the caption of figure 11 as per suggestion. The changes are highlighted with red text.

Comment 3: Correct "Parkinson's" in Line 67?

Response 3: The word is changed and corrected.

Comment 4: Rewrite the legend for Figure 2?

Response 4: Corrected as per suggestion.

Comment 5: Move the abbreviations after the chemical names in the legend of Figure 3?

Response 5: I have revised the caption of figure 3 as per suggestion.

Comment 6: Describe how the compounds were purified in Line 113?

Response 6: I have elaborated the method of purification as per suggestion citing a reference too. All of these changes were highlighted with red text.

Comment 7: Change "drugs" to "compounds" or "agents" in the title (Line 2)?

Response 7: Revised the title as per suggestion.

Comment 8: Change the "0.0" to "-" in Table 3.

Response 8: Corrected.

……………………………..Thank you for appreciable and valuable comments……………………….

Round 2

Reviewer 3 Report

Thanks for the revision.